

# Preoperative bedside test indicators as predictors of difficult video laryngoscopy in obese patients: a prospective observational study

Liumei Li[*], Guanyu Yang[*], ShiYing Li, Xue Liu, Ya Fei Zhu and Qinjun Chu

Department of Anesthesiology and Perioperative Medicine, Zhengzhou Central Hospital Affiliated to Zhengzhou University, Zhengzhou, Henan, China
[*] These authors contributed equally to this work.

## ABSTRACT

**Purpose.** The aim of this study was to identify factors associated with difficult video laryngoscopy in obese patients.

**Methods.** A total of 579 obese patients undergoing elective laparoscopic weight loss surgery were intubated with a single-lumen endotracheal tube using a video laryngoscopy under general anesthesia, and the patients were divided into two groups based on the Cormack-Lehane classification (difficult video laryngoscopy defined as ≥ 3): the easy video laryngoscopy group and the difficult video laryngoscopy group. Record the general condition of the patient, bedside testing indicators related to the airway, Cormack-Lehane classification during intubation, and intubation failure rate.

**Results.** The findings of this study show that the incidence of difficult video laryngoscopy in obese patients is 4.8%. Multivariate logistic regression analysis indicated that body mass index was significantly associated with difficult video laryngoscopy (OR = 1.082, 95% CI [1.033–1.132], $P < 0.001$).

**Conclusion.** For Chinese obese patients without known difficult airways, the incidence of difficult video laryngoscopy is 4.8%. Body mass index is associated factors for the occurrence of difficult video laryngoscopy, with an increased risk observed as body mass index rise.

## INTRODUCTION

With the improvement of living standards, the proportion of obese people in China has rapidly increased in the past forty years, and therefore the proportion of obese patients undergoing surgery has also been continuously rising (*Chen et al., 2023*). For obese patients who require general anesthesia for surgery, anesthesiologists usually choose tracheal intubation. Airway management in obese patients is relatively challenging because of the accumulation of fat deposits in the airway, thickening of the soft palate and tongue, resulting in a narrow pharynx and severely affecting the exposure of the glottis; in addition, tongue posterior displacement is prone to occur after anesthesia induction, which also

Corresponding author
Qinjun Chu, jimmynetchu@163.com

affects the exposure of the glottis (*Lin et al., 2022*; *Anderson & Shashaty, 2021*; *Prathep et al., 2022*). Obese patients typically have increased oxygen consumption and concomitant restrictive lung disease, which reduces the tolerance to failed attempts at tracheal intubation (*Jones & Nzekwu, 2006*; *Bhardwaj, Garg & Devgan, 2019*; *Mahul et al., 2016*).

One study has shown that the failure rate of tracheal intubation in obese patients can be as high as 15–20% (*Higgs et al., 2018*). The laryngoscope is the most commonly used tool for tracheal intubation, and video laryngoscopy has gradually replaced direct laryngoscopy due to its absolute advantages (*Hyman et al., 2021*; *Cooper, 2018*; *Wu, 2018*). Good vocal cord exposure is the key to success when performing tracheal intubation using a laryngoscope. Previous studies have identified various factors that contribute to difficult laryngoscopy in obese patients, such as body mass index (BMI), Mallampati grade, and neck circumference (*Wang, Sun & Huang, 2018*; *Mashour et al., 2008*; *Riad et al., 2016*). However, there is limited research on the factors related to difficult video laryngoscopy in obese patients. Therefore, it is important to identify these factors in advance in order to adequately prepare for potential difficult airway. For this purpose, we evaluated the relationship between preoperative bedside testing indicators and the occurrence of difficult video laryngoscopy in obese patients.

## METHODS

This is a single-center, prospective, observational study approved by the Ethics Committee of Zhengzhou Central Hospital on March 18, 2022 (reference number 202236) and registered on March 29, 2022 in the Chinese Clinical Trial Registry (ChiCTR2200058090). All patients had signed a written informed consent before protocol enrollment, in keeping with the Helsinki Declaration.

Recruitment of patients scheduled for laparoscopic weight loss surgery under general anesthesia was from April 8, 2022 to February 28, 2023. Inclusion criteria: age $\geq$18 years, BMI $\geq$28 kg/m$^2$, American Society of Anesthesiologists (ASA) grade II-III, single-lumen endotracheal intubation using a video laryngoscope. Exclusion criteria: surgery cancelled for various reasons, known difficult airway, mental or neurological disorders, patient refusal to participate.

General information about the patient was collected, including age, weight, height, BMI, sex, comorbidities (diabetes mellitus, rheumatology or rheumatoid arthritis, cervical spondylopathy, neck scar contracture), ASA grades, and dentition (missing, denture). A trained researcher measured the patient's interincisor distance, thyromental distance, neck circumference, assessed the patient's Mallampati grade and neck movement the day before surgery. This researcher did not participate in the subsequent study process and the results were kept confidential. Table 1 shows the definitions of concepts such as interincisor distance, thyromental distance, neck circumference, Mallampati grade, neck movement, and Cormack-Lehane (C-L) classification, in order to minimize bias as much as possible.

On the day before surgery, all patients underwent a routine preoperative visit by a resident anesthesiologist and signed an informed consent for anesthesia. Perioperative management of the patients was jointly carried out by the resident anesthesiologist and an

**Table 1** Definition of measurement indices.

| Index | Definition |
|---|---|
| Interincisor distance, cm | The distance between the upper and lower central incisors at the maximum opening of the mouth (inter-gingival distance in edentulous patients) |
| Thyromental distance, cm | The distance from the notch of thyroid cartilage to the inner mentum when the neck is fully extended |
| Neck circumference, cm | Horizontal circumference of neck through thyroid cartilage |
| Mallampati grades (Open the mouth and extend the tongue as far as possible without pronouncing) | I: Soft palate, pharyngopalatine arch, uvula and hard palate visible |
| | II: Soft palate, uvula and hard palate visible |
| | III: Soft and hard palate visible |
| | IV: Only the hard palate visible |
| Neck movement | Flexion and extension 35 to 45 degrees, left and right rotation 60 to 80 degrees, left and right lateral bending about 45 degrees |
| C-L classification | I: The entire vocal cords are visible |
| | II: Parts of the vocal cords are visible |
| | III: The epiglottis is visible, but the vocal cords are not |
| | IV: Epiglottis is not even visible |

**Notes.**
C-L, Cormack-Lehane.

attending anesthesiologist. The resident anesthesiologists all had two years of experience in anesthetic management and they had performed approximately 2,000 tracheal intubations using the UE video laryngoscope (TD-C-IV; UE Medical Corporation, Zhejiang, China) in the last two years. The attending anesthesiologist possesses more than 5 years of professional experience. Upon entering the operating room, all patients received standard monitoring (electrocardiography, non-invasive blood pressure, and pulse oximetry). Anesthesia induction was performed using sufentanil 0.5 ug/kg (total body weight), propofol 2–2.5 mg/kg (ideal body weight), and rocuronium 0.6–1 mg/kg (ideal body weight). After 5 min, endotracheal intubation was performed by the resident anesthesiologist. The patient's head is positioned in the sniffing position and tracheal intubation was performed using as well as a UE video laryngoscope fitted with a Macintosh size 3 laryngoscope blade. The preferred endotracheal tube sizes are 7.0# for females and 7.5# for males (with various sizes available). For patients with a C-L grade of ≥3 during intubation, the following strategies should be attempted in sequence: BURP maneuver (*Knill, 1993*), size 4 laryngoscope blade, and a combination of BURP maneuver with a size 4 laryngoscope blade. Alternative options for failed intubation under video laryngoscopy include fiberoptic bronchoscope guidance, laryngeal mask airway, cricothyroid membrane puncture, and tracheostomy. The anesthesiologist performing the intubation records the C-L grade during the video laryngoscopy, whether the BURP maneuver and size 4 laryngoscope blade were used, the success of intubation under video laryngoscopy, and the rescue measures taken after failed intubation under video laryngoscopy on the case report form. Based on the C-L grade (≥3 defined as difficult video laryngoscopy), patients were divided into two groups: the easy
video laryngoscopy group (group E, C-L grade I-II) and the difficult video laryngoscopy group (group D, C-L grade III-IV).

SPSS 23.0 software was used for data statistical analysis. Normal distribution metric data is represented by mean $\pm$ standard deviation, and independent sample $t$-test is used for between-group comparisons. Non-normal distribution metric data is represented by median and interquartile range, and the Mann–Whitney $U$ test was used for between-group comparisons. Count data is represented as cases (%), and between-group comparisons were performed using chi-square test or Fisher's exact test. Univariate analysis is conducted on potential variables related to the difficult video laryngoscopy. After exclusion of multicollinearity, variables with $P < 0.05$ and those of clinical significance were included in the multivariate logistic regression analysis. $P < 0.05$ indicates statistically significant differences.

# RESULTS

The study included a total of 608 patients who underwent elective laparoscopic weight loss surgery under general anesthesia. 18 patients were excluded for not meeting the inclusion criteria, and three patients had their surgery cancelled for various reasons. There were two patients with mental and neurological disorders, and six patients refused to participate. Finally, a total of 579 patients were included in the analysis (Fig. 1). The general characteristics of the two groups of patients are shown in Table 2.

Among the 579 patients, 551 (95.2%) were in group E and 28 (4.8%) were in group D. The success rate of intubation was 100% in both groups. Among the 28 patients in group D who had difficult video laryngoscopy, 11 were successfully intubated using the BURP maneuver with a size 3 laryngoscope blade, 10 were successfully intubated using a size 4 laryngoscope blade, and seven were successfully intubated using the combination of the BURP maneuver and a size 4 laryngoscope blade.

Univariate analysis, as shown in Table 2, identified weight, height, BMI, sex, ASA grade, and neck circumference as factors significantly associated with difficult video laryngoscopy ($P < 0.05$). Although Mallampati grade did not show statistical significance in univariate analysis, it is commonly used as a clinical indicator for assessing difficult airways. Considering its clinical significance, it was also included in the multiple logistic regression analysis. The three factors of weight, height and BMI were only included as one factor, BMI, due to multicollinearity; ASA grade was not included as it is an assessment of the patient's overall risk and does not correlate well with the airway; neck circumference is the circumference of the neck at the level of the cricoid cartilage and does not reflect the distribution of neck fat, and several studies have shown that neck circumference correlates poorly with difficult laryngoscopies in obese patients (*Özdilek et al., 2018*; *Eiamcharoenwit et al., 2017*; *Siriussawakul et al., 2016*), therefore neck circumference was also not included. Three factors, BMI, sex, and Mallampati grade, were included in the final multiple logistic regression analysis.

The results of the multiple logistic regression analysis, as shown in Table 2, indicate that BMI was significantly associated with difficult video laryngoscopy in obese patients (OR =1.082, 95% CI [1.033–1.132], $P < 0.001$).
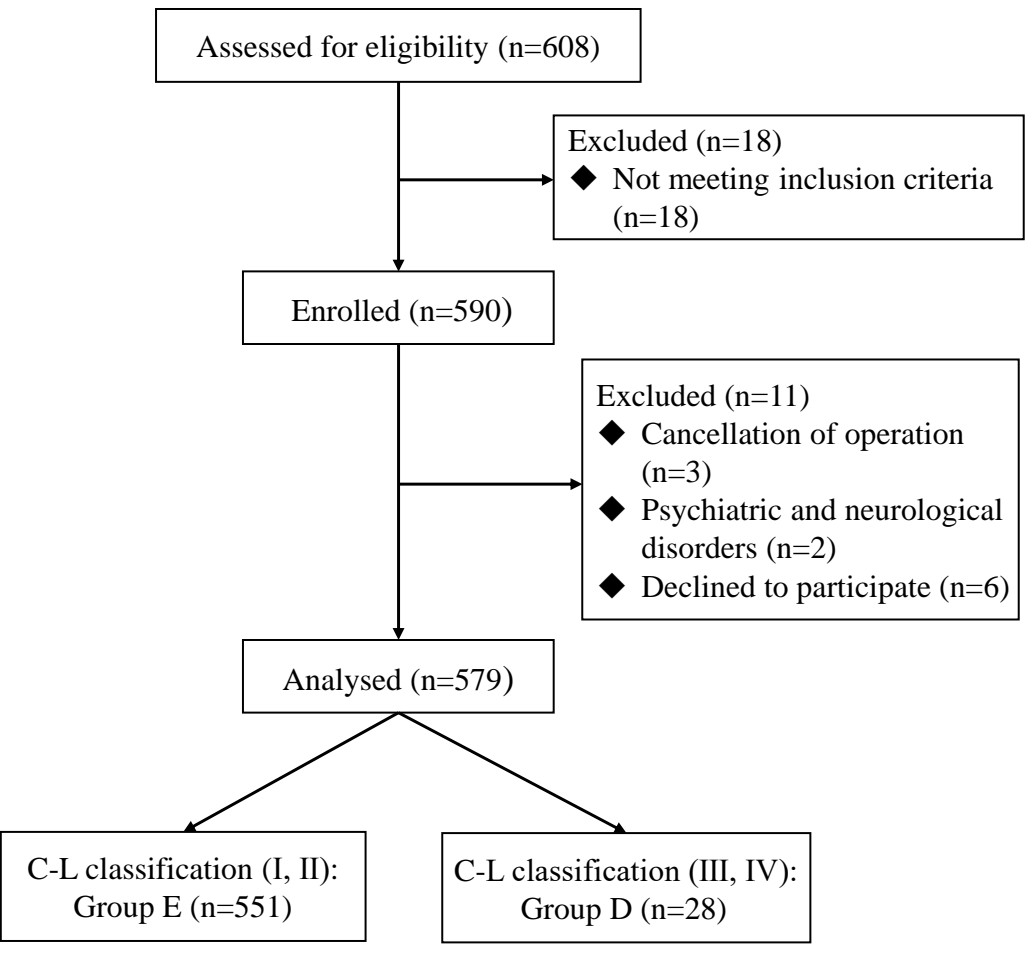

**Figure 1** Flow diagram of patient recruitment.

## DISCUSSION

The results of this study show that the incidence of difficult video laryngoscopy in obese patients is 4.8%. BMI was found to be a correlated factor for difficult video laryngoscopy in obese patients, with an increased risk observed as BMI increased.

In a study by *Prathep et al. (2022)*, the incidence of difficult laryngoscopy in morbidly obese patients was reported to be as high as 14.8%, whereas in this study, the incidence of difficult video laryngoscopy in obese patients was 4.8%. This difference may be primarily attributed to the use of a video laryngoscopy in this study, which significantly improves the exposure of the glottis during intubation compared to direct laryngoscopy.

In this study, difficult video laryngoscopy in obese patients can be resolved through the BURP maneuver and the use of larger laryngoscope blades. Therefore, it is important for anesthesiologists to master the correct BURP maneuver and have larger-sized laryngoscope blades available. If intubation fails under video laryngoscopy, tools such as a laryngeal mask

**Table 2  Patient characteristics, univariate analysis, and multivariate regression analysis.**

| Variables | Univariate analysis | | | Multivariate analysis | |
|---|---|---|---|---|---|
| | E (n = 551) | D (n = 28) | P | OR (95% CI) | P |
| Age, years | 33 ± 8 | 34 ± 8 | 0.396 | | |
| Weight, kg | 109 ± 24 | 131 ± 25 | <0.001 | | |
| Height, cm | 166.4 ± 8.5 | 171.1 ± 10.3 | 0.006 | | |
| BMI, kg/m$^2$ | 39.3 ± 6.8 | 44.9 ± 8.9 | <0.001 | 1.082 (1.033–1.132) | <0.001 |
| Sex, n (%) | | | 0.005 | 0.472 (0.212–1.050) | 0.066 |
| Male | 174 (31.6) | 16 (57.1) | | | |
| Female | 377 (68.4) | 12 (42.9) | | | |
| Comorbidities, n (%) | | | | | |
| Diabetes mellitus | 175 (31.8) | 13 (46.4) | 0.106 | | |
| Rheumatology or rheumatoid arthritis | 0 (0) | 0 (0) | / | | |
| Cervical spondylopathy | 11 (2.0) | 1 (3.6) | 0.452 | | |
| Neck scar contracture | 6 (1.1) | 0 (0) | >0.999 | | |
| ASA grades, n (%) | | | <0.001 | | |
| II | 233 (42.3) | 2 (7.1) | | | |
| III | 318 (57.7) | 26 (92.9) | | | |
| Dentition, n (%) | | | | | |
| Missing | 5 (0.9) | 0 (0) | >0.999 | | |
| Denture | 18 (3.3) | 2 (7.1) | 0.251 | | |
| Interincisor distance, cm | 4.4 ± 0.5 | 4.3 ± 0.6 | 0.420 | | |
| Thyromental distance, cm | 7.9 ± 1.0 | 8.2 ± 1.2 | 0.286 | | |
| Neck circumference, cm | 41.4 ± 4.5 | 44.7 ± 5.2 | <0.001 | | |
| Mallampati grades, n (%) | | | 0.477 | | |
| I | 175 (31.8) | 7 (25.0) | | | |
| II[*] | 102 (18.5) | 3 (10.7) | | 0.621 (0.153–2.518) | 0.505 |
| III[*] | 258 (46.8) | 17 (60.7) | | 1.443 (0.574–3.626) | 0.436 |
| IV[*] | 16 (2.9) | 1 (3.6) | | 1.197 (0.136–10.567) | 0.872 |
| Neck movement, n(%) | | | 0.094 | | |
| Normal | 550 (99.8) | 27 (96.4) | | | |
| Limited | 1 (0.2) | 1 (3.6) | | | |

**Notes.**
In the multivariate analysis, the P-value of the Hosmer-Lemeshow goodness-of-fit test was 0.572; the variance inflation factors for BMI, sex, and Mallampati classification were 1.086, 1.097, and 1.015, respectively, indicating that there was no multicollinearity among the three variables.
Data shown by mean ± SD or numbers (%).
BMI, body mass index; ASA, American Society of Anesthesiologists.
*In the multivariate analysis, compared to Mallampati I.

or fiberoptic bronchoscope guidance can be used, and if necessary, an invasive airway can be established to ensure ventilation.

Multiple research studies have shown that BMI is a relevant factor in difficult laryngoscopy for obese patients (*Wang, Sun & Huang, 2018*; *Yuan et al., 2024*; *Juvin et al., 2003*). The findings of this study also indicate that BMI is a relevant factor in difficult video laryngoscopy for obese patients, with an increased risk as BMI increases. Therefore, in clinical practice, it is important to be vigilant about the occurrence of difficult airways

in patients with higher BMI and to be adequately prepared. For example, we can try using conscious tracheal intubation or preparing various airway related tools (such as supraglottic devices, fiberoptic bronchoscopy, *etc.*) to minimize the risk of difficult airway occurrence.

ASA grade is a rating system developed by the American Society of Anesthesiologists to assess the physical condition and surgical risk of patients. Although the *P*-value of ASA grading was <0.05 in the univariate analysis, it was ultimately not included in the subsequent multivariate analyses, considering on the one hand its weak specificity with respect to airway assessment, and on the other hand the fact that there were fewer variables that could have been included in multivariate regression analyses in this study. There are also no studies showing that ASA grading predicts the occurrence of difficult video laryngoscopy.

Mallampati grading, which assesses the airway by observing the oral structures, is a commonly used method by anesthesiologists to determine the presence of potential difficult airways. However, its effectiveness has shown inconsistent results in various studies. Multiple research findings have indicated a correlation between Mallampati grades III-IV and difficult laryngoscopy in normal patients (*Wu et al., 2021*; *Tamire, Demelash & Admasu, 2019*; *Harjai et al., 2023*). Our previously conducted research, which is currently being published, revealed a significant association between Mallampati grades III-IV and the occurrence of difficult video laryngoscopy in normal patients. *Lee et al. (2015)* showed a weak correlation between Mallampati grading and C-L grading in obese patients, suggesting that it cannot predict difficult laryngoscopy in this population. Additionally, *Mashour et al. (2008)* research revealed that Mallampati grading has low sensitivity and predictive value when used in obese patients. Our study similarly concluded that Mallampati grading cannot be used to predict difficult video laryngoscopy in obese patients, highlighting the need for anesthesiologists to rely on other more reliable indicators rather than Mallampati grading when assessing the presence of potential difficult airways in obese patients. Even when faced with a patient classified as Mallampati grade I, preparation for managing a difficult airway must be undertaken.

Neck circumference had a *P*-value of <0.05 in the univariate analyses, but again it was not included in the subsequent multivariate analyses, mainly because multiple research results show that neck circumference has limited predictive value for difficult laryngoscopy in obese patients (*Özdilek et al., 2018*; *Eiamcharoenwit et al., 2017*; *Siriussawakul et al., 2016*). This may be because neck circumference does not represent the distribution of neck fat.

In the study by *Özdilek et al. (2018)*, the incidence of difficult laryngoscopy in obese male and female patients was 10.8% and 4.8% respectively. In this study, the incidence of difficult video laryngoscopy in obese male and female patients was 2.8% and 2.1% respectively, which may be related to the use of a video laryngoscope in this study, which improved the exposure of the glottis. However, the results of this study show that sex cannot predict the occurrence of difficult video laryngoscopy in obese patients.

This study has the following limitations: This is a study conducted on the obese population in China, and the differences between different racial groups may affect the generalizability of the results of this study; This study only used one type of video

laryngoscope, which may affect the generalizability of the results of the study; The C-L classification is suitable for direct laryngoscopy, and its application to video laryngoscopy may not be appropriate, and it may be necessary to establish grading criteria that match the exposure of the video laryngoscope in the future; Finally, difficult intubation includes difficult laryngoscopy, which can only indicate unclear glottic exposure and has limited predictive value for difficult intubation.

## CONCLUSION

For Chinese obese patients without known difficult airways, the incidence of difficult video laryngoscopy is 4.8%. BMI is an associated factor for the occurrence of difficult video laryngoscopy, with an increased risk observed as BMI rise.

## ACKNOWLEDGEMENTS

We express our gratitude to the bariatric surgery department for their support in facilitating our patient recruitment process.

### Funding
The authors received no funding for this work.

### Competing Interests
The authors declare there are no competing interests.

### Author Contributions
- Liumei Li conceived and designed the experiments, performed the experiments, prepared figures and/or tables, authored or reviewed drafts of the article, and approved the final draft.
- Guanyu Yang conceived and designed the experiments, analyzed the data, prepared figures and/or tables, authored or reviewed drafts of the article, and approved the final draft.
- ShiYing Li performed the experiments, authored or reviewed drafts of the article, and approved the final draft.
- Xue Liu performed the experiments, authored or reviewed drafts of the article, and approved the final draft.
- Ya Fei Zhu performed the experiments, authored or reviewed drafts of the article, and approved the final draft.
- Qinjun Chu conceived and designed the experiments, analyzed the data, prepared figures and/or tables, authored or reviewed drafts of the article, and approved the final draft.

### Human Ethics
The following information was supplied relating to ethical approvals (*i.e.,* approving body and any reference numbers):

The Ethics Committee of Zhengzhou Central Hospital, (reference number 202236).

## Clinical Trial Ethics

The following information was supplied relating to ethical approvals (*i.e.,* approving body and any reference numbers):

The Chinese Clinical Trial Registry, (ChiCTR2200058090).

## Data Availability

The raw data are available in the Supplementary File.

## Clinical Trial Registration

The following information was supplied regarding Clinical Trial registration:

ChiCTR2200058090

## Supplemental Information

Supplemental information for this article can be found online at http://dx.doi.org/10.7717/peerj.17838#supplemental-information.

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
