# Peer review of "Preoperative bedside test indicators as predictors of difficult video laryngoscopy in obese patients: a prospective observational study"

_PeerJ, doi:10.7717/peerj.17838_

## Round 0.1 · original submission · Major Revisions

The reviewers have highlighted several areas for improvement in your manuscript, with particular reference to experimental design and statistics. Please take care to address each point in detail when resubmitting your manuscript.

Reviewer 1 ·

Basic reporting

The rationale of the study is clear regarding the predictors of difficult video laryngoscopy in obese patients which the knowledge in this field is still low. However, there were many points of study design which need to be addressed including the statistical analysis.

The relevant results were still not clear when the methodology part was still not clear.

The English language is acceptable.

Experimental design

The ethical section is clear as well as the trial registration. However, there were many points of study design which need to be addressed including the statistical analysis as below.

1. Definition of obesity: Since the authors used BMI >= 28 kg/m2 which represented only mild obesity. Is there any proportion or number of patients who had morbid obesity (BMI> 35 or > 40 kg/m2) since the mild obesity might relate with lower incidence of difficult intubation (DI) as well.

2.The intubation performers were by the residents. How many residents performed intubations? Which year of the anesthesia residents performed the intubation since different years of experience could affect the success rate and laryngoscopic view? Is there any difference in experience among them?
- Regarding UE scope, how long was the residents trained to use this device since the UE screen is quite small compared to the glidescope.
- Regarding the definition of difficult intubation, was there any criteria of difficult intubation beside CL >=3 such as attempts of intubation, since the experience of performers during applying the VDL also affect the easy/ difficult view.

3. A little bit confusion in the methodology section regarding when to perform Mcintosh size 3 or 4 (line 86, 87) during intubation whether before or after applying video laryngoscopy.

Statistical analysis
1. How did the authors select the variables from univariate to the initial multivariate model?
2. The authors mentioned using conditional multiple logistic regression analysis (line 127) which is usually used in Matched case-control study design not prospective cohort study. If so, what is the matching variables? Could the authors explain more why using conditional logistic regression instead of simple logistic regression.
3. The sample size calculation must be performed since the outcome (DI case) was few (28 cases). How many risk factors did the authors expect to accomplish?

Validity of the findings

The validity of the finding can not be verified since the methodology part was still not clear.

Additional comments

Discussion section
- Since the sample size of DI was low (28 cases), the only two predictors could be found. Is there any modifiable predictor that could be done to minimize risk of DI in obese patient besides using video laryngoscope?
- What is the implications of the study?

References section
- There were only two references that recently published after 2020, one already mentioned in the introduction (Prathep 2022). Therefore, is there any more update references after 2020 could be cited in the discussion section?

Reviewer 2 ·

Basic reporting

This study offers valuable insights into what predicts difficult video laryngoscopy in obese patients, pinpointing BMI and ASA grade as key factors.

Experimental design

The multivariate logistic regression analysis is missing some details about the model's assumptions checks, such as multicollinearity diagnostics and goodness-of-fit tests. Please include these to make sure the findings are robust.

The confidence interval for the ASA grade (1.245-27.000) is very wide, showing potential instability in the model. Consider discussing why this wide interval might occur, like sample size limits or data variability.

Some terms and phrases need to be clearer. For example, "BMI has a statistically significant impact" should be "BMI is significantly associated with." Ensure all terms are consistent with standard medical and statistical language.

The study mentions BMI and ASA grade are important for predicting difficult video laryngoscopy but doesn’t explain enough how this is relevant in clinic or what interventions might be done based on these findings. Expanding on these points will make the study more useful.
Conclusion Validation:

Validity of the findings

The conclusion says BMI and ASA grade are predictors of difficult video laryngoscopy, but this claim needs more validation. Consider discussing how these findings fit with other research and any limitations that might affect how generalizable the results are.

---

## Round 0.2 · accepted · Accept

The changes made have adequately addressed the reviewers comments, and accordingly, the manuscript is ready for publication.

Reviewer 2 ·

Basic reporting

No

Experimental design

No

Validity of the findings

No